# Psychosocial Interventions to Improve Psychological, Social and Physical Wellbeing in Family Members Affected by an Adult Relative’s Substance Use: A Systematic Search and Review of the Evidence

**DOI:** 10.3390/ijerph18041793

**Published:** 2021-02-12

**Authors:** Ruth McGovern, Debbie Smart, Hayley Alderson, Vera Araújo-Soares, Jamie Brown, Penny Buykx, Vivienne Evans, Kate Fleming, Matt Hickman, John Macleod, Petra Meier, Eileen Kaner

**Affiliations:** 1Population Health Sciences Institute, Newcastle University, Newcastle upon Tyne NE2 4AX, UK; deborah.smart@ncl.ac.uk (D.S.); hayley.alderson@ncl.ac.uk (H.A.); eileen.kaner@ncl.ac.uk (E.K.); 2Faculty of Behavioural, Management and Social Science, University of Twente, 7522 Enschede, The Netherlands; vera.araujo-soares@utwente.nl; 3Institute of Epidemiology & Health, University College London, London WC1E 6BT, UK; jamie.brown@ucl.ac.uk; 4School of Humanities and Social Science, University of Newcastle, Callaghan 2308, Australia; Penny.Buykx@newcastle.edu.au; 5School of Health and Related Research, The University of Sheffield, Sheffield S10 2TN, UK; 6Adfam, 120 Cromer Street, London WC1H 8BS, UK; v.evans@adfam.org.uk; 7Public Health Policy and Systems, Liverpool Centre for Addiction Research, University of Liverpool, Liverpool L69 3BX, UK; Kate.Fleming@liverpool.ac.uk; 8Population Health Sciences Institute, University of Bristol, Bristol BS8 1TL, UK; Matthew.Hickman@bristol.ac.uk (M.H.); John.MacLeod@bristol.ac.uk (J.M.); 9Institute of Health & Wellbeing, University of Glasgow, Glasgow G12 8QQ, UK; Petra.Meier@glasgow.ac.uk

**Keywords:** substance use, family, affected other, psychosocial intervention, systematic review

## Abstract

It is estimated that over 100 million people worldwide are affected by the substance use of a close relative and often experience related adverse health and social outcomes. There is a growing body of literature evaluating psychosocial interventions intended to reduce these adverse outcomes. We searched the international literature, using rigorous systematic methods to search and review the evidence for effective interventions to improve the wellbeing of family members affected by the substance use of an adult relative. We synthesised the evidence narratively by intervention type, in line with the systematic search and review approach. Sixty-five papers (from 58 unique trials) meeting our inclusion criteria were identified. Behavioural interventions delivered conjointly with the substance user and the affected family members were found to be effective in improving the social wellbeing of family members (reducing intimate partner violence, enhancing relationship satisfaction and stability and family functioning). Affected adult family members may derive psychological benefit from an adjacent individually focused therapeutic intervention component. No interventions fully addressed the complex multidimensional adversities experienced by many families affected by substance use. Further research is needed to determine the effect of a multi-component psychosocial intervention, which seeks to support both the substance user and the affected family member.

## 1. Introduction

Alcohol and drug use are common in families [1] and represent a major public health concern [2]. It is estimated that over 100 million people worldwide are affected by a family member’s substance use [3] and suffer significant stress, akin to trauma, which causes psychological, social, and physical problems [3,4]. Whilst worry for the substance-using relative is a defining feature of the impact upon the family [3], the effects can be severe and long lasting [5] including increased physical [6] and psychological health problems [6,7], resulting in high service usage [7]. The multi-dimensional impact of substance use is often aggravated by additional related stressors including interpersonal conflict, financial problems, burden of care, and family disharmony [8,9]. Furthermore, there is established evidence of the association between substance use and intimate partner violence and abuse [10], both in the perpetrator [11] and the victim [12], as well as a correlation between heavy use of substances and the severity of violence [13]. Children are highly vulnerable to the effects of a family member’s substance use, particularly when the substance user is their parent, with evidence of an association with a wide range of harms [14] including abuse and neglect [15]. Children whose parents have a substance use disorder are more likely to suffer an injury [16] and experience health problems [17] than children whose parents do not use substances. Pre-school children are at risk of delays in cognitive and language development [18], and lower educational attainment [19], resulting in poor life chances [17] and they have a high prevalence of substance use disorders [14].

### Study Objectives

Despite clear evidence of harm to close family members, substance use has primarily been viewed from an individualistic perspective, with cause, effect, and intervention research focusing upon the person who uses substances [20]. There has, however, been a growing acknowledgement of the importance of involving family members in the treatment of substance users [21,22], underpinned by assumptions that the family may in some way be part of the problem, offer part of the solution, or, more recently, in acknowledgement that affected family members may benefit from treatment “in their own right” [22]. We aimed to systematically search and review the international published evidence for psychosocial interventions for family members affected by an adult relative’s substance use to determine the type of interventions used and what is known about the impact of the interventions at improving their psychological, social, and physical wellbeing. In line with the systematic search and review approach, our review scope was broad to enable us to achieve a best evidence synthesis as well as identifying gaps in the knowledge base [23].

## 2. Methods

The international literature was searched using electronic databases Medline (OVID), PsycINFO (OVID), CINAHL (EBSCO), SCOPUS, International Bibliography of Social Science (ProQuest), ProQuest Criminal Justice (ProQuest), ProQuest Social Science Journals (ProQuest), ProQuest Sociology (ProQuest), Social Service Abstracts (ProQuest), and Sociological Abstracts (ProQuest). A search strategy using key terms, thesaurus headings, Boolean, and proximity operators was adapted for each database and implemented. Key terms related to the population (e.g., family, significant other, affected other); condition (e.g., alcohol use/abuse/dependency/disorder; drug use/abuse/dependency/disorder); intervention (e.g., family therapy, systemic therapy, conjoint therapy, couples therapy); and study type (e.g., trial, randomized control trial, clinical trial). Full details of the search strategy are available within the Appendix A. All databases were searched from inception to November 2020 and no language or geographical restrictions were applied. Two researchers independently screened all titles and abstracts using pre-specified inclusion and exclusion criteria, retrieving full papers for all potentially eligible studies and evaluating in full text. Discrepancies at each stage were resolved by discussion or by consulting a third researcher if consensus could not be reached.

### Review Inclusion Criteria

Studies utilising a trial design (randomised controlled trials, controlled trials, randomised trials, and quasi-experimental trials) were included if they measured the effectiveness of psychosocial interventions upon the psychological, social, or physical wellbeing of family members affected by a close adult relative’s substance use. We included trials which used diagnostic and validated tools to measure the relative’s substance use (illicit drug and alcohol use), as well as those which relied upon the affected family members’ report that the relative’s substance use was a concern. Psychological wellbeing included reduced stress, coping, anxiety, or more generic psychological and mental health problems. We defined social wellbeing as measures relating to family and relationships including reduced conflict and violence, improved family communication, and measures of child welfare (e.g., abuse/neglect and change in legal status). Physical wellbeing included reduced physical health symptoms, disease, and health compromising behaviours (e.g., smoking, drug use, diet, and exercise). We defined psychosocial interventions as any non-pharmacological intervention. These include but are not limited to systemic family and couples therapy (approaches involving the family/partner to address problems in their relationships and interactions; helping family members to better understand one another, change negative behaviours, and resolve conflicts); unilateral family interventions (teaching strategies and skills to influence the substance-using relative’s behaviour and motivate change, with the primary aim of increasing treatment seeking behaviour in the substance user); and psychological interventions which recognise that the substance use of a relative can negatively impact family members. The affected family member is the only recipient of the intervention, which typically seeks to enhance their ability to cope, alleviate stress, or address trauma (e.g., forgiveness therapy, stress-strain-coping-support model or five step approach). We included a variety of comparison conditions. This included control conditions of no intervention, wait-list control, alternative active intervention, or interventions which engaged only the focal substance user.

Papers were excluded if: they reported upon smoking or caffeine use only, the substance user was below the age of 18 years, the affected family member was not a direct recipient of at least one of the psychosocial interventions, or if the intervention effect upon the affected family member was not reported in the paper. The methodological quality of each study included was assessed according to the criteria presented in the Cochrane risk of bias tool [24]. This domain-based tool addresses seven domains: random sequence generation and allocation concealment (selection bias); blinding of participants and providers (performance bias); blinding of outcome assessor (detection bias); incomplete outcome data (attrition bias); selective outcome reporting (reporting bias); and other sources of bias. A judgement relating to the risk of bias for each entry is assigned in terms of low, high, or unclear risk. The results of our review are narratively synthesised, grouped according to the intervention type, and organised by outcome. This analytical approach allowed us to consider the effectiveness of a broad range of interventions, which, due to the heterogeneous nature of the outcome tools used, prevented meta-analysis.

## 3. Results

### 3.1. Description of Studies

We identified 65 papers (reporting on 58 unique trials) which met our inclusion criteria. The majority (*n* = 34, 59%) of the trials were conducted in the US; five in Iran; four in Australia; three each in the UK and Sweden; two each in Germany and Spain; and one each from Brazil, Korea, Vietnam, the Netherlands, and Mexico. The sample sizes of the included studies ranged from 12 to 325 individuals, dyads, or families, depending upon the level of intervention (mean sample 103, with a combined total of 5955). As it is not possible to blind intervention recipients to a psychosocial intervention, all of the trials were assessed as being of high risk of selection bias. As the majority also utilised self-report outcome tools, these trials were also at a high risk of performance bias. Additionally, a number of trials did not use random assignment. The psychosocial interventions were delivered to a range of different affected family members: the partner/spouse of the substance user (*n* = 30), parents and/or children (*n* = 14) which included child aged <18 years (*n*= 3), adult child (*n* = 5), and family (usually a combination of multiple relationship types, *n* = 14). Of these interventions, 25 trials reported on interventions where a single affected family member was the only recipient of an intervention; 2 trials reported on an intervention which was primarily delivered to the affected family member alone, but they were also given the option to attend a minimal number of sessions conjointly with the substance user; 31 trials reported findings of interventions where the affected family member(s) received a conjoint intervention with the substance user against a control condition. Most of the trials reported on mainly female samples. Thirty-two of the trials reported on psychosocial interventions where the substance-using relative used alcohol only, 11 papers intervened in relation to drug use, and 14 papers included both alcohol and/or drug use. Trials most commonly reported on social outcomes for the affected family member (*n* = 34 trials), followed by psychological outcomes (*n* = 27 trials). Health outcomes were rarely reported (*n* = 8 trials). Figure 1 provides further details of the flow of the studies identified for the review. Table 1 provides an overview of the nature (indicated in table by shape) and size (indicated in table by size of the shape) of the literature with an evidence and gap map. A summary of findings is detailed in Table 2.

### 3.2. Parental and Child Interventions (15 Trials)

Fifteen trials intervened to address the impact of parental substance use disorder upon the family. The trials varied in who received the intervention and the mechanisms by which they sought to affect change. There were broadly two types of interventions for parents and children affected by parental substance use—those that intervened with the parent to enhance parent skill and those that intervened with the child to address the impact that their parent’s substance use has had upon them. We identified five trials which intervened with the substance-using parent to enhance their parenting skill and family functioning. In these trials, the affected non-using partner and/or child were typically involved to support change within the substance-using parent’s behaviour, for example, to provide a means to practice the newly acquired skill, or to reinforce positive behaviour.

Despite this indirect focus, the interventions examined in these trials were mostly found to be effective at improving family functioning and relationships. At 6 months follow-up, children whose families received parenting skill intervention in addition to methadone treatment (opioid maintenance/detoxification therapy for opioid use disorder) reported significantly more parental involvement and activities with their parent than those children whose parents received standard methadone treatment [25]. One further trial, which combined behavioural couples’ therapy (BCT) with parent skills training for parents in families where the father has an alcohol use disorder, reported significant improvement on parental discipline scales (laxness and over-reactivity) when compared to individual therapy and BCT alone at both 6 and 12 months follow-up [26]. This suggested that parenting was achieved through the addition of parental skills training. Additionally, family therapy has been found to improve family relationships, parental involvement, family communication, family bonds, and family cohesion within families where there is parental substance use disorder [27]. However, a further trial that examined systems therapy in families where a parent has an alcohol use disorder found that this intervention did not significantly improve marital satisfaction or family satisfaction more than the comparison condition of a problem solving approach at 6 months follow-up [28].

A minority of these trials also measured child outcomes, showing some evidence of effect on a small number of very diverse outcomes. Family therapy was found to reduce child maltreatment potential in families where a parent has a substance use disorder at 6 and 10 months follow-up, demonstrating medium effect sizes compared to standard treatment [29]. Behavioural couples’ therapy, provided to parents with either an alcohol or drug use disorder in conjunction with their non-using partners, was found to result in significantly greater psychosocial adjustment in the children at 6 and 12 months when compared to those children whose parents received individual behavioural therapy (IBT) or couples psychoeducational control treatment (PACT) [30]. Two trials that examined the substance use outcomes of children showed mixed results. One trial examining a parenting skill intervention found there was a reduction in alcohol or marijuana use disorder in male children of substance-using parents but not female children [31]. A further trial of family therapy found children aged 8–16 years who participated in family therapy with their substance-using mother had a short-term reduction in alcohol and tobacco. However, at follow-up these children were found to have higher levels of alcohol and tobacco than those children whose mother received a health education intervention [27].

Eight trials examined the effectiveness of interventions which directly intervened with the children of substance-using parents, with a primary focus of addressing the impact their parent’s substance use had upon them. Of these trials, three intervened with dependent age children and five intervened with adult children. The trials examining interventions for dependent age children included a diverse range of interventions, showing mixed and low-quality evidence of effect. One trial randomly assigned children aged 8–12 years old whose parent had a substance use disorder to a psychoeducational intervention or non-educational play and fun sessions. The psychoeducational intervention was not found to result in better mental wellbeing, coping, self-perceived autonomy, or parent–child relationship than non-educational play, with both groups making improvements [32]. Evidence of a reduction in participant anger was reported at 2 months post treatment in a trial of emotional intelligence group training as compared to wait-list control for adolescent boys of fathers with an opiate use disorder [33]. Group assertiveness training for female adolescent children (aged 12–15 years) where both parents have an opiate use disorder reported significant improvements in happiness and assertiveness compared to wait-list control at 6 weeks post treatment [34]. Three trials considered the effectiveness of self-help interventions to improve the wellbeing of adult children of alcohol-using parents. When comparing Al-anon group intervention to education classes, improvements were observed on depression at 6 months follow-up [35]. Similarly, a group accessing self-help intervention resulted in more improvements in depressive symptoms than both a group receiving a psychotherapy intervention and those receiving no intervention [36]. A trial of computer-based self-help intervention found the experimental intervention improved levels of depression and self-acceptance more than the therapy-only group [37]. Two further trials examined interventions for adult children of alcohol-using parents—one examining the effectiveness of forgiveness therapy compared to conflict resolution [38] and another comparing a coping programme to an alcohol programme and a group combining both interventions [39]; however, neither found between-group differences on outcomes measured.

### 3.3. Behavioural Couples and Family Therapy (19 Trials)

Behavioural couples’ therapy (BCT) typically consisted of a combination of individual drug and alcohol treatment sessions for the substance user and conjoint behavioural couples counselling sessions totalling between 10 and 32 sessions. Within the conjoint sessions, couples were typically encouraged to discuss how the non-using partner can positively support the using partner and were taught how to communicate more effectively and increase positive behavioural exchanges. In additional to examining the impact of the intervention upon the substance user’s alcohol or drug consumption, these trials focused primarily upon the impact of the intervention upon the relationship including incidence of intimate partner violence, conflictual communication, and overall relationship satisfaction. Of the five trials that examined the effectiveness of the intervention at reducing intimate partner violence, four reported reductions. Two trials providing BCT to couples where the male partner used either alcohol or drugs [40] or drugs only [41] reported greater reduction in male to female violence when compared to individual behavioural therapy. In both of these trials, the incidence of violence in the BCT group was around half that of the comparison group at 12 months follow-up, whilst a further trial, which examined the effectiveness of BCT where the female partner used alcohol or drugs, reported significantly reduced male to female acts of violence at 12 month follow-up [42]. The proportion of couples reporting conflictual communication reduced from pre to post treatment with those in couples therapy, whilst conflict in couples where the alcohol-using male received individual therapy increased slightly [43]. In a trial of BCT plus parent skills training for alcohol-using parents, Lam et al. (2009) reported that both the BCT participants and BCT with parent training made clinically meaningful reductions in percentage of days of any violence with effect sizes of small to medium, whilst the individual behavioural therapy did not [26]. Two further trials reported the intervention did not result in greater improvements than the comparison condition—one comparing alcohol-focused couples’ therapy to non-directive supportive counselling for male alcohol users [44], and one examining BCT compared to individual behavioural therapy for women who use alcohol [45].

The majority (91%) of trials which measured the effectiveness of BCT for increasing the level of relationship satisfaction reported significant results when compared to individual therapy. These results were found in partners of drug and/or alcohol users at 3-month [46], 4-month [47], 6-month [46,48,49,50], 12-month [26,30,42,51,52,53], and 18-month follow-up [54]. In addition, partners reported significantly fewer thoughts of separation and divorce from their alcohol-using partner [48] and a lower percentage of days separated from their drug using partner during the follow-up period than controls [51]. Whilst a further trial reported significant improvements between pre and post treatment for partners of alcohol users receiving BCT compared to non-directive supportive counselling, these results had dissipated by 12 months follow-up [44]. In two trials which compared BCT with additional relapse prevention sessions to standard BCT, the authors reported that relapse prevention further increased relationship satisfaction in partners of alcohol users [55,56]. A trial of BCT for male alcohol users, compared to a control of an individual peer intervention, reported no between-group difference in relationship satisfaction at 12, 18, and 24 months follow-up [57]. No between-group effects upon relationship satisfaction were found in trials of BCT versus non-conjoint therapy in partners of alcohol users [45,57,58,59] and alcohol and/or drug users [60].

Three trials examined the effectiveness of BCT upon the psychological wellbeing of the non-using partner. Two trials found that BCT for partners of alcohol users did not improve partner psychological wellbeing significantly more than the control condition [44,61], whilst one trial found that partners who received BCT with their alcohol-using partner had improved psychological wellbeing at 18 months follow-up [54].

### 3.4. Systemic Family Interventions (6 Trials)

Systemic family therapy typically treats the substance users and their family as a system, recognizing the role of the family in the development and treatment of the adult relative’s substance use. It is generally assumed that positive changes in the family system will result in positive changes in the relative’s substance use. Most of the trials examining the effectiveness of systemic family therapy at improving the wellbeing of the affected family member found significant effects. In the only trial of family therapy not to include the focal user, family members of drug users reported significant reductions in co-dependency in the family therapy group compared to counselling only at post intervention and 90 days post intervention [62]. A trial examining the effectiveness of a long-term family intervention (9–18-month intervention period), for a population dually diagnosed with schizophrenia and substance use disorder, found that participating relatives reported significant improvement in mental health at follow-up 3 years post randomization [63]. Family functioning was found to be significantly improved at 3 and 6 months follow-up in the families of injecting drug users who received 4–6 sessions of family therapy intervention as compared to standard care. Whilst coping and depressive symptoms were found to have improved at 3 months, only the effect for depressive symptoms was maintained at 6 months follow-up [64]. A family-centred empowerment model was found to improve social support and quality of life in family members of methamphetamine users post intervention, when compared to no intervention [65]. No between-group effects upon relationship satisfaction for partners of alcohol users were found in a trial of marital systems therapy versus non-conjoint therapy [66].

### 3.5. Unilateral Family Interventions (7 Trials)

Unilateral family interventions intervene with the affected family member in order to teach them strategies by which they can support or influence the substance-using family member to address their substance use. The interventions typically include education on the process of change, on how to effectively place pressure upon substance user to action change as well as support to address the affected family member’s behaviour which may enable the relative’s substance use. The primary outcome of unilateral family interventions is, therefore, typically the substance use and/or treatment seeking behaviour of the substance user. We identified a number of trials of unilateral family interventions which also examined the affected family member outcomes. Whilst these trials were found to consistently improve the treatment seeking behaviour and treatment engagement of the substance-using relative, these trials rarely reported improved wellbeing in the affected other. In trials examining various versions of the Pressures to Change approach for family members affected by an adult relative’s alcohol use, no significant differences were found between groups on depression [67,68], personal problems [69], life satisfaction, partner distress [68], and marital discord [68,69]. However, one trial found Pressures to Change improved marriage satisfaction significantly more than the comparison groups of Al-anon and wait-list control [67].

Four trials measured the effectiveness of community reinforcement and family training (CRAFT) and did not demonstrate superiority over comparison interventions for outcomes relating to the wellbeing of the affected family member. CRAFT for family members of alcohol users [70] and drug users [71] were found to result in significant improvements in the mental health of the affected family member [70,71], family cohesion, and problems [70,71]; however, there were no between group differences when compared to Al-anon [70,71] and a Johnson Institute Intervention (confrontational family intervention) [70]. One trial comparing CRAFT to wait-list control for the family members of alcohol users found the intervention improved relationship satisfaction but the effect upon the affected relatives’ mental health was mixed [72]. In contrast, a further trial which compared an internet-based version of CRAFT to wait-list control for the families of alcohol users found no between-group difference in the affected family members’ mental health at follow-up [73].

### 3.6. Psychosocial Interventions for the Individual Affected Family Member (11 Trials)

Psychosocial interventions for the family members of a substance user are non-pharmacological therapeutic interventions delivered to individuals or groups, which seek to tackle the psychological, social, and personal problems experienced by the affected family member. Trials examining the effectiveness of coping skills interventions for the family members of alcohol users mostly did not find the intervention was effective at improving the affected family member’s wellbeing. In a single trial, coping skills training was found to reduce the incidence of partner physical violence at 6 months follow-up and levels of depression at 12 months follow-up compared to wait-list control [74]. However, two further trials found that coping skills training—one with the family members of alcohol and/or drug users [75,76] and one with the partners of alcohol users [77,78]—did not result in greater improvements in coping [75,76,77,78], psychological wellbeing [77,78], or physical symptoms [75,76] at follow-up points between 3 and 24 months when compared to a manualised self-help intervention [75,76], group-based coping, or standard information [77,78].

Seven trials examined behavioural interventions provided to the family members of alcohol and drug users in order to address interactional problems, enhancing the affected family member’s assertiveness and communication skills. Five of these trials reported improvements in the others’ affected psychological wellbeing for one or more of the reported outcomes. This related to anxiety [79,80], depression [81], self-esteem [82], coping [82], co-dependent behaviour [83], and partner enabling behaviour [80]; one trial reported a reduction in the burden of care for family members of drug users [84]. Conversely, no improvement was reported relating to anxiety in family members of alcohol users [81] or mental health problems in families of drug users [85]. Two trials found behavioural interventions did not improve relationship quality [79,81] or reduce family conflict in family members affected by an adult relative’s alcohol use [79].

One trial which examined the effectiveness of forgiveness group therapy in wives of alcohol users found that the therapy led to improvements in resilience and self-esteem at 3 months follow-up [86].

## 4. Discussion

We identified an emerging body of literature examining interventions to improve the wellbeing of family members affected by an adult relative’s substance use—signalling a shift away from a largely individualistic perspective upon the focal substance user, to one which includes or focuses upon the wider family. This is welcomed and is indicative of an increasing awareness of the importance of the family within the context of substance use. A wide variety of interventions were identified, with differing mechanisms of impact and outcomes. The findings of our systematic search and review suggest that interventions which integrate substance use treatment for the user with a family focused component may bring about social benefit for the affected family members. These interventions introduce behavioural strategies to improve family functioning and reduce conflict whilst placing emphasis upon teaching the affected adult family members how to support the substance user to reduce or abstain from alcohol or drugs. They are commonly delivered to the substance user and their significant other conjointly, in dyads or family groups. Previous systematic reviews of BCT [87,88,89] have found the intervention to be effective at reducing drug and alcohol use. However, these reviews were primarily focused upon the substance user, with little examination of family outcomes. The summative conclusions of our systematic search and review provide tentative evidence that integrated behavioural interventions for substance-affected families, which situate efforts to reduce or abstain within a family context, may also result in positive outcomes, particularly relating to intimate partner violence, relationship adjustment, and family functioning. There was an absence of evidence that these interventions may positively affect the psychological wellbeing of affected family members however, with none of the trials that examined behavioural family interventions measuring family member mental health outcomes.

Our systematic search and review found a body of trials which examined interventions that were provided directly and solely to the affected other. These interventions often sought to remove the worry (by affecting change in the substance use) or teach the affected family member how to “worry better” through behavioural strategies or skills training. Unilateral family interventions such as CRAFT and Pressure to Change sought to teach families members how to influence their relative’s substance use and treatment-seeking behaviour. Whilst unilateral family interventions were consistently found to be effective at enabling family members to influence the substance-using relative to reduce their use or increase treatment engagement, there was insufficient evidence that this brought psychological or social relief in the affected family members. It is clear that whilst the origins of the stressors may intertwine with the substance use, increasing treatment engagement for the user is not enough to benefit the affected others. Whilst this may alleviate acute worry, it does not do anything to address the long lasting impact of the trauma. Similarly, those interventions which sought to improve the affected users’ ability to cope with the impact of the using relative’s behaviour (without changing the stressor) were insufficient to result in significantly better outcomes for the adult or child family members affected by the substance use of an adult relative. Our review identified some examples of behavioural interventions delivered to the affected family member alone, and that may be effective at improving mental health and wellbeing. These interventions were typically based upon interventions which have previously been found to be effective in general populations for depression and anxiety such as cognitive behavioural therapy [90], or those specifically designed to alleviate stress and enhance skills and had been enhanced by explicit consideration of how the substance use of their relative affects them and their behaviours.

Despite the well documented traumatic impacts substance use is likely to have upon significant others [3,4,9], our search found only a small minority of trials that examined interventions provided directly to the family member in their own right and which specifically sought to intervene to improve family members’ psychological outcomes. The majority of trials of interventions which are delivered exclusively or jointly to affected family members maintain a focus upon reducing the substance use. When families are involved in treatment, this is most frequently as a means of affecting change in the focal user. These interventions do not go far enough in responding to the needs of the family and the multi-dimensional impact of substance use, including the long lasting impact upon the psychological and social wellbeing of the family. Substance use by an adult relative has a substantial impact upon the family. It is clear that addressing substance use is an important first step in the recovery of the family. Conjoint behavioural family therapy may offer promise in improving the social wellbeing and functioning of the family. However, it is likely that affected family members will also need support to recover from the impact. This suggests that both the substance user and the affected other may benefit from additional individual therapeutic sessions, in order to respond to their individual as well as their interpersonal needs. Such an approach would provide the substance-using relative with drug and alcohol treatment and the affected family member with mental health treatment to address the psychological impact, whilst jointly developing behavioural strategies for substance use and its multi-dimensional impact that must be addressed by the intervention.

## 5. Limitations

Within this review, we have examined all psychosocial interventions for any family member affected by an adult relative substance misuse. Whilst this broad focus is appropriate for the aims of this systematic search and review, this approach limits our ability to draw confident conclusions about the effectiveness of specific interventions at improving specified outcomes, within defined populations. It is important to acknowledge that the nature of the relationship between the family member and the substance misuser is likely to result in differential effects—impacts which may be more or less malleable to change. Our review provides an evidence overview inclusive of summary findings of effect, which introduces the opportunity for future systematic review and meta-analysis of focused review questions as well as evidence which may be used to further investigate promising intervention developments through empirical study.

There are also a number of limitations within the literature. Whilst the studies typically utilised randomised trial design, the strength of evidence is greatly reduced by a high risk of allocation and performance bias and small sample sizes. The trials are typically pilot trials and as such, are not sufficiently powered to conduct reliable statistical testing or cost effectiveness analysis. The results reported in the trials are, therefore, at risk of both type I and type II error, wherein the null hypothesis is either incorrectly rejected or retained. Furthermore, the trials often compared the experimental intervention to active interventions, many of which have an evidence base within adult substance-using populations. The use of active and, on some occasions, highly intensive comparison interventions is likely to reduce the ability of the trial to identify significant effects achieved by the intervention. The majority of the trials were conducted in the USA and those trials that reported on the ethnicity of the participants described a majority white sample (76%). Whilst the common core elements of the affected family member experience have been found across and within cultures, important variation exists [91]. That said, the remaining trials (*n* = 25) were conducted in eleven different countries, including some trials from low to middle income countries which is important in advancing understanding of cross-cultural variation [21]. Structural subordination and dependence, wherein family members are reliant financially or socially upon the substance-using relative, deepens the strain of a relative’s substance use [91]. As such, the greater number of studies focusing upon female family members is reflective of this. Nonetheless, the paucity of trials examining interventions for male affected family members is a gap within the literature. Although included in the non-specified family member group, no studies specifically examined the effectiveness of interventions for parents of adult substance users and only one trial examined interventions for same sex couples [52]. 

## 6. Conclusions

There is a large volume of literature examining psychosocial interventions which have been found to improve the psychological and social wellbeing of family members affected by an adult relative’s substance use. However, these interventions do not go far enough to address the needs often experienced within substance-affected families. There is a need for research which develops and evaluates interventions which seek to address the complex multidimensional adversities experienced by many families affected by substance use. Further research is needed to determine the effect of a multi-component psychosocial intervention, which seeks to support both the substance user and the affected family member, with equal focus upon their needs.

## Figures and Tables

**Figure 1 ijerph-18-01793-f001:**
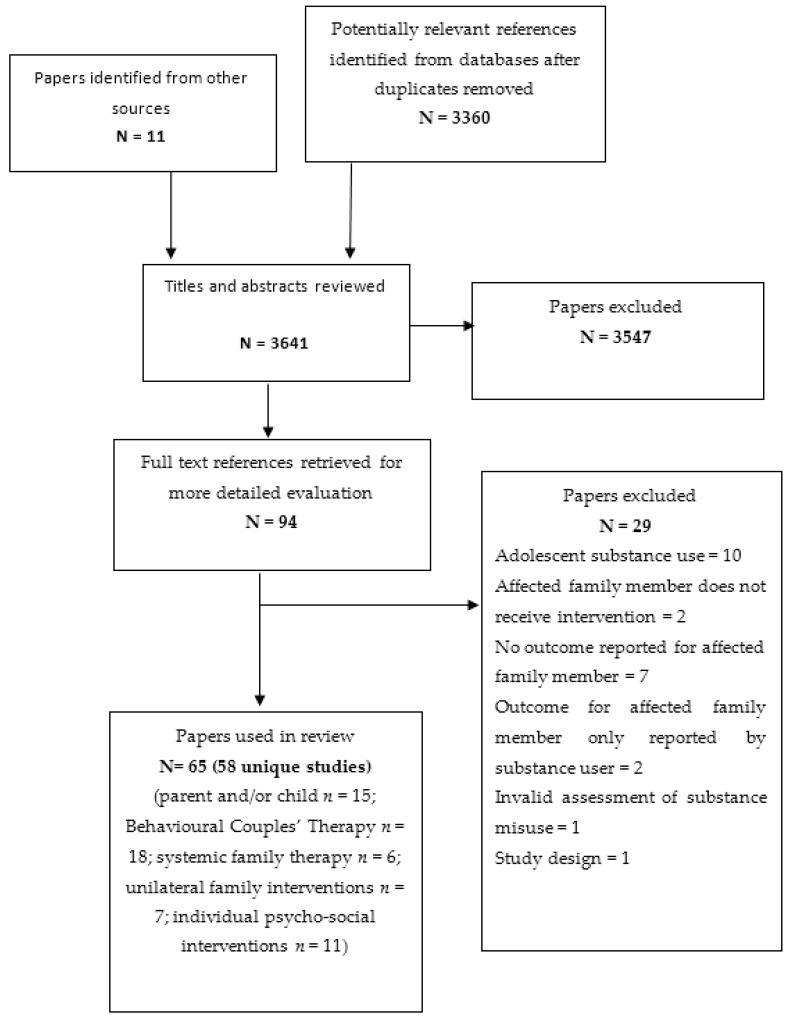
Flow of studies.

**Table 1 ijerph-18-01793-t001:** Evidence and gap map.

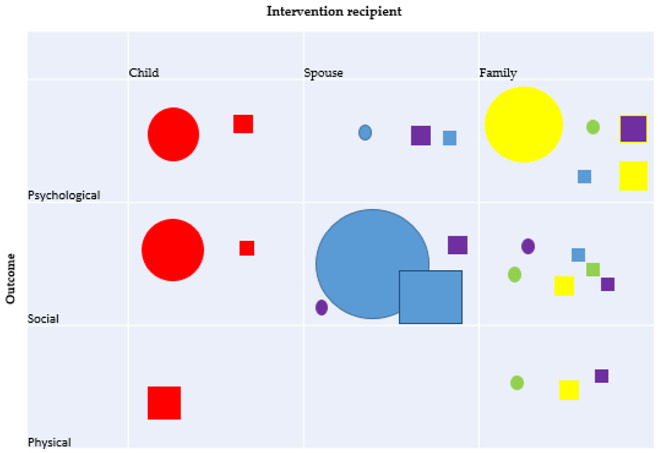
**Parent/child interventions**	Evidence of effect	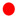	Evidence of no effect/adverse effect	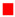
**Behavioural couples’ therapy**	Evidence of effect	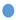	Evidence of no effect/adverse effect	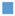
**Systemic family therapy**	Evidence of effect	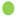	Evidence of no effect/adverse effect	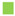
**Unilateral family intervention**	Evidence of effect	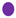	Evidence of no effect/adverse effect	
**Psychosocial interventions**	Evidence of effect	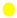	Evidence of no effect/adverse effect	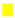

**Table 2 ijerph-18-01793-t002:** Evidence of effectiveness from included studies.

Author, Date, Country	Participants	Intervention	Comparison	Experimental Intervention Outcome	Risk of Bias Summary
**Parent and Child Intervention**					
*Parental alcohol use and dependent age children*					
Bennun (1988) *UK*	*n* = 12 families; alcohol misusing parents and children; mean age 16 years	Milan systems therapy *	Problem-solving treatment based on social learning theory *	Non-Significant (NS) effect on Marital Adjustment Test (MAT) and Family Satisfaction Rating (FS) at 6 months.	Unclear
Lam (2009) *USA*	*n* = 30 families; females and their male alcohol misusing partners, with at least one child aged 8–12 years	(1) Parent skills plus BCT (PSBCT); (2) BCT; 24 sessions each	IBT; 24 sessions	PSBCT-improved parental style (Parenting Scale-PS) and both PSBCT and BCT-reduced spousal violence using Timeline Followback Spouse Violence (TLFB-SV) and improved spousal relationship satisfaction on the Dyadic Adjustment Scale (DAS) at 6 and 12 months follow-up.	Low
*Parental alcohol use and adult children*					
Hansson (2006) *Sweden*	*n* = 82; adult children of parents with alcohol problems; mean age 25.6 years; 61% female alcohol	(1) Coping (CBT) only; (2) Coping and alcohol intervention *	Alcohol intervention focusing upon the adult child’s drinking only *	NS on Short Index of Problems (SIP), Symptom Checklist-90 item (SCL-90), Interview Schedule for Social Interaction (ISSI), questionnaire coping with parents’ abuse at 12 months follow-up.	Low
Gustafson (2012) *USA*	*n* = 23; adult children of alcoholic parents, aged 18 years and older; 63% female	(1) Computer-based intervention only; (2) Computer-based intervention plus group therapy *	Group therapy only *	No significance test conducted due to small sample size. Self-constructed questionnaire measuring responsibility and blame, depression, anxiety, loneliness, personal growth, positive relations, and self-acceptance completed post intervention.	High
Kingree (2000) *USA*	*n* = 114; adult children of alcoholics, mean age 34.5 years; 32% female	Al-anon (family support groups); 12 meetings	Substance abuse education classes *	Improved perceived status benefits (self-constructed questionnaire) and depression (CES-D) at 6 months post treatment	Unclear
Kuhns (1997) *USA*	*n* = 64; adult children of alcoholic parents, mean age 20.4 years	(1) Self-help; (2) Psychotherapy; 8 sessions	No intervention	Both self-help and psychotherapy reduced depression on Centre for Epidemiological Studies Depression Scale (CES-D) 3 months post treatment	Unclear
Osterndorf (2011) *USA*	*n* = 12; adult children of alcoholics, 91.7% female	Forgiveness therapy; 12 sessions	Conflict resolution; 12 sessions	NS on Enright Forgiveness Inventory (EFI); Anxiety on State-Trait Anxiety Inventory (STAI); anger on State-Trait Anger expression Inventory (STAXI); Beck Depression Inventory (BDI-II), Positive Relations with Others (PRO) at 3-month follow-up.	Unclear
*Parental drug use and dependent age children*					
Catalano (1999)*;* Haggerty (2008) *USA*	*n* = 130 families; parents in receipt of methadone; 75% female; child aged 3–14 years	Focus on Families; parent skill training and case management; 33 sessions	Methadone treatment *	Higher parental involvement (self-constructed questionnaire) at 6-month follow-up. Adolescent males less likely to have an alcohol or marijuana use disorder; NS for females (Composite International Diagnostic Interview—CIDI) at 12–14-year follow-up.	Low
Hojjat (2016) *Iran*	*n* = 57; female children with both parents being dependent upon opium, aged 12–15 years	Group assertiveness training; 8 sessions	Wait-list control	Increased happiness (Oxford Happiness Scale) and assertiveness (Gambrill–Richey Assertion Inventory) one month post intervention.	High
Hojjat (2017) *Iran*	*n* = 57; male children aged 14–18 years who have an opium dependent father	Emotional intelligence training; 8 sessions	No intervention	Reduced anger (STAXI-(2) at 4-month follow-up.	High
*Parental alcohol and/or drug use and dependent age children*					
Donohue (2014)	*n* = 72 substance abusing mothers referred to child welfare for neglect, and their children	Family behavioural therapy * plus standard treatment	Standard treatment *	Significant reductions in child maltreatment potential at 6 and 10 months follow-up.	Low
Kelley (2002) *USA*	*n* = 135 families; all female partners of substance abusing male, with at least one child aged 6–16 years	(1) BCT; (2) Individual behavioural therapy (IBT); 32 sessions each	Couples based psychoeducational attention control (PACT); 32 sessions	Improved child psychosocial functioning (Pediatric Symptom Checklist—PSC) and relationship satisfaction (DAS) at 6 (in both alcohol and drug using group) and 12 months (drug using group).	Low
Orte (2008) *Spain*	*n* = 93; children of dependent substance misusing parents; mean age 10.6 years	Family competence programme; 14 sessions	Standard outpatient drug treatment *	Improved family relationships, parental involvement, family communication, family bonds, and family cohesion (Strengthening Families Programme validated instruments) at 3-month follow-up	High
Bröning (2019) *Germany*	*n* = 218; children of substance misusing parents, aged 8–12 years; 47.7% female; participants excluded if in receipt of additional intervention	Psych-educational intervention; 9 sessions	Non educational play and fun sessions *	NS on coping measures (adapted version Stress and Stress Management in childhood and adolescence—SSKJ); self-efficacy (Generalised Self-Efficacy Scale); self-concept (Self-Perception Profile for Children, child version—SPPC); health-related quality of life (KIDSCREEN-27); self-constructed questionnaires on parent relationship, mental distress, and social isolation at 6-month follow-up.	Low
Bartle-Haring (2018) *USA*	*n* = 183 families; children of substance-using mothers, mean age 11.54 years	Ecologically based family therapy; 12 sessions	Women’s health education; 12 sessions	Short term immediate decrease (6 months) in alcohol and tobacco followed by an increase (12 and 18 months). NS for cannabis (Timeline Followback—90 item; TLFB-90).	Unclear
**Behavioural couples and family therapy**					
*Interventions for partners/families of an adult who uses alcohol*					
Fals-Stewart (2005) *USA*	*n* = 60 dyads; female partners of male alcohol misusers	(1) Brief BCT; 18 sessions; (2) standard BCT; 24 sessions	(1) Couples psychoeducational attention control (PACT); (2) IBT; 18 sessions each	Brief and standard BCT significantly improved relationship satisfaction (DAS); however, brief BCT was not equivalent to standard at 12-month follow-up.	Low
Fals-Stewart (2006) *USA*	*n* = 138 dyads; male partners of female alcohol misusers	(1) BCT; (2) IBT; 32 sessions each	PACT; 32 sessions	BCT group less acts of violence TLFB-SV and improved relationship (DAS) at 12-month follow-up.	Low
Fals-Stewart (2009b) *USA*	*n* = 100 dyads; same sex couples where one partner has alcohol use disorder	BCT; 32 sessions	IBT; 32 sessions	Increased relationship satisfaction (DAS) throughout follow-up period (3, 6, 9, and 12 months).	Low
Halford (2001) *Australia*	*n* = 44 dyads; female partners of alcohol abusing males	(1) Stress management; (2) Alcohol focused couples’ therapy; 15 sessions each	Non-directive counselling; 15 sessions	NS results for stress and burden (Relative Stress Scale and General Health Questionnaire—GHQ); relationship satisfaction (DAS and Martial Status Inventory—MSI); and physical aggression within relationship (Conflict Tactics Scale—CTS) at 6-month follow-up.	High
Kuenzler (2003) *USA*	*n*= 50 dyads; partners also drank alcohol; 70% above recommended levels	BCT *	Family systems therapy *	NS for partners’ psychological wellbeing as measured by BDI post treatment.	High
Lam (2009) *USA*	*n* = 30 families; females and their male alcohol misusing partners, with at least one child aged 8–12 years	(1) Parent skills plus BCT (PSBCT); (2) BCT; 24 sessions each	IBT; 24 sessions	Both PSBCT and BCT reduced spousal violence (TLFB-SV) and improved spousal relationship satisfaction (DAS) at 6 and 12 months follow-up.	Low
McCrady (1986); McCrady (1991) *USA*	*n* = 47 dyads; partners of alcohol abusers	(1) Alcohol-focused couple involvement (AFSI); (2) Alcohol behavioural martial therapy (ABMT) *	Minimal spouse involvement *	Increased relationship satisfaction (MAT) and psychological status (Psychosocial Functioning Inventory) in ABMT at 6 months follow-up.	Low
O’Farrell (1985); O’Farrell (1992) *USA*	*n*= 34 dyads; female spouses of male alcoholics	(1) Couples attended mutual support group; (2) BCT; 10 sessions each	Individual peer counselling *	BCT significantly higher relationship satisfaction (MAT) and higher martial stability (Martial Stability Inventory—MSI) at 6 and 12 months.	Low
O’Farrell (1993)*;* O’Farrell (1997) *USA*	*n* = 59 dyads; female partners of male alcoholics	BCT and relapse prevention; 25 sessions	BCT; 10 sessions	Higher relationship satisfaction (MAT) at 12 months effect not found at 18 or 24 months follow-up.	Low
O’Farrell (2016) *USA*	*n* = 101 dyads; partners of alcohol dependent patients; 29.7% female	Group BCT; 23 sessions	Standard BCT; 23 sessions	Group BCT significantly lower relationship satisfaction (DAS) than standard BCT at 3 and 6 months follow-up.	Low
Schumm (2014); *USA*	*n* = 105 dyads; male partners of alcohol dependent females	BCT; 26 sessions	IBT; 26 sessions	NS on relationship satisfaction (DAS and Relationship Happiness Scale -RHS) or intimate partner violence (CTS) at 12 months follow-up.	Low
Vedel (2008) *Netherlands*	*n* = 64 dyads; partners (male and female) of alcohol misusers	BCT; 10 sessions	IBT; 10 sessions	NS on relationship satisfaction (Maudsley Martial Questionnaire) at 6 months follow-up.	Low
Walitzer (2004) *USA*	*n* = 64 dyads; female partners of problem drinkers	(1) Couples alcohol focused; (2) Couples alcohol focused and BCT; 10 sessions	Individual focused; 10 sessions	NS on relationship satisfaction (DAS) or spouse support measures (Partner Interaction Questionnaire and Significant-other Behaviour Questionnaire) at 6 and 12 months follow-up.	Unclear
Walitzer (2013) *USA*	*n* = 64 dyads; female partners of problem drinkers	(1) Couples alcohol focused (CAF); (2) CAF and BCT; 10 sessions	Individual focused; 10 sessions	Decrease in conflictual communication in two couple-involved groups (Rapid Martial Interaction Coding System) post treatment.	Low
*Interventions for partners/families of an adult who uses drugs*					
Fals-Stewart (2001) *USA*	*n* = 36 dyads; female partners of men in receipt of methadone	BCT; 24 sessions	IBT; 24 sessions	Increased relationships satisfaction (DAS and MHS) post treatment.	Low
O’Farrell (2017) *USA*	*n* = 61 dyads; male partners of female drug users; 45% of males were also drug users	BCT; 26 sessions	IBT; 26 sessions	NS on relationship satisfaction (DAS), lower days separated (% of days) at 12 months follow-up.	Low
*Interventions for partners/families of an adult who uses alcohol and/or drugs*					
Fals-Stewart (2002) *USA*	*n* = 80 dyads; female partners of male substance misusers	BCT; 34 sessions	IBT; 22 sessions	Less male to female acts of violence (CTS) at 12 months follow-up.	Low
Fals-Stewart (2009a) *USA*	*n* = 207 dyads; female partners of substance misusing males	BCT; 32 sessions	IBT; 32 sessions	Lower male to female violence (TLFB-SV) at 12 months follow-up.	Low
O’Farrell (2010) *USA*	*n* = 29 dyads; family members other than spouses of substance misusing relative	Behavioural family therapy; 24 sessions	IBT; 24 sessions	NS on relationship satisfaction (RHS) at 3 and 6 months follow-up.	Low
Winters (2002) *USA*	*n* = 75 dyads; male partners of female substance misusers	BCT; mean 39.5 sessions	IBT; mean 38.4 sessions	Increased relationships satisfaction (DAS and RHS) at 12 months follow-up.	Unclear
**Systemic family therapy**					
*Interventions for families of an adult who uses alcohol*					
McKay (1993) *USA*	*n* = 51; family of alcohol misusers	Conjoint systemic therapy *	Non conjoint alcohol coping *	NS difference in family functioning as measured on the Family Assessment Device (FAD) at 6 months follow-up.	High
Zweben (1988) *USA*	*n* = 116 dyads; partners (male and female) of alcohol abusers	Couples martial therapy *	Couples advice counselling *	NS on Revised Martial Relationship Scale (RMRS), DAS, and Edmonds Martial Conventionality Scale (EMC) at 6, 12, and 18 months follow-up.	Low
*Interventions for families of an adult who uses drugs*					
Ahmad-Abadi (2017) *Iran*	*n* = 61; co-dependent partners of drug users	Communication family therapy; 10 sessions	Counselling; 1–3 sessions	Reduced co-dependency as measured on the Holyoake co-dependency index (HCI) at 3 months follow-up.	Unclear
Ghasemi (2014) *Iran*	*n* = 285; male and female family of methamphetamine users	Family empowerment model *	No intervention	Improved quality of life as measured on the Short Form Health Survey-36 post treatment.	Unclear
Li (2014) *Vietnam*	*n* = 83; family members of IV drug users; 100% female	Family and user sessions delivered separately; 6 sessions	Standard care *	Improved coping (brief COPE scale) at 3 months only, depression (Zung Self-Rating Depression Scale) at 6-month follow-up only and family functioning (adapted from Family Functioning Scale) at 6 and 12 months.	Low
*Interventions for families of an adult who uses alcohol and/or drugs*					
Mueser (2012) *USA*	*n* = 108; family of dually diagnosed patients	Family intervention * (18-month intervention period)	Education for family *	Improved knowledge (self-reported), mental health (as measured by SF-12), worry, and stigma (Family Experiences Interview Schedule—FEIS) at 36-month follow-up.	Low
**Unilateral family interventions**					
*Interventions for families of an adult who uses alcohol*					
Barber (1995) *Australia*	*n* = 23; partners of heavy drinkers; 96% female	(1) Pressures to Change (individual); (2) Pressures to Change (group) *	Wait-list	NS on Martial consensus scale (MCS), Life Satisfaction Scale (LSS), personal problems (self-report) post treatment and 3 months later.	Unclear
Barber (1996) *Australia*	*n*= 48; mostly partners (daughters and mothers also) of heavy drinkers, 94% female	(1) Pressures to Change (individual); (2) Pressures to Change (group) *	(1) Wait-list; (2) Al-anon (family support group) *	Individual-improved marriage discord-measured Drinkers Partner Distress Scale (DPDS), personal problems (self-report). NS on depression (DPDS) post treatment.	Unclear
Barber (1998) *Australia*	*n* = 38, female partners of male heavy drinkers	(1) Pressures to Change (individual); (2) Pressures to Change (self-help manual) *	(1) Wait-list control	Individual-improved marriage satisfaction (DPDS), NS on life satisfaction (LSS) only when combined pressures to change groups decreased depression (DPDS) 1 month post treatment.	Unclear
Bischof (2016) *Germany*	*n* = 78; majority female (97.9%) and partners (79.1%), of alcohol-dependent relatives	CRAFT *	Wait-list control *	Improvements in mental health on Mental Health Inventory but not on BDI or SCL-90. Improvements in and relationship satisfaction (RHS) at 12 months follow-up.	Low
Eek (2020) *Sweden*	*n* = 94; majority female (92.3%) and partners (86.2%), of mostly alcohol-dependent relatives (94.7%)	iCRAFT; 10–12 weekly sessions	Wait-list control	All mental health outcomes were NS at 12 and 24 weeks follow-up (as measured by Montgomery–Åsberg Depression Rating Scale-Self Assessment (MADRS-S) and Depression, Anxiety and Stress Scale (DASS)).	Unclear
Miller (1999) *USA*	*n* = 130 family members of alcohol users	(1) CRAFT; 12 sessions; (2) Johnson Institute Intervention; 6 sessions	Al-anon; 12 sessions	All outcomes relating to affected family member were NS (as measured on BDI, State-Trait Anxiety Inventory, STAXI, Spouse Enabling Inventory and Spouse Influence Inventory, self-esteem scale (self-reported) and physical symptoms scale (self-reported) Family Environment Scale (FES), DAS, and (RHS) at 3, 6, 9, and 12 months follow-up.	Low
*Interventions for families of an adult who uses drugs*					
Kirby (1999) *USA*	*n* = 32; family members of drug users; 94.4% female	CRAFT; 14 sessions	Al-anon; 10 sessions	All outcomes relating to affected family member were NS (as measured on Family Impact Survey (FIS), Profile of Mood States (POMS), Social Adjustment Scale, and Self-Esteem Scale) post treatment.	Unclear
**Psychosocial interventions for the individual affected family member**					
*Interventions for families of an adult who uses alcohol*					
Copello (2009); Velleman (2011) *UK*	*n* = 136 family members of substance misusers	Stress-strain-coping-support model; 1 session	Self-help manual with similar content	NS on two validated tools measuring physical and psychological coping (Symptom Rating Test and Coping Questionnaire) at 12 months follow-up.	Low
Cruz-Almanza (2006) *Mexico*	*n* = 35; female partners of alcohol misusing men	Rational-Emotive Behavioural Therapy; 18 sessions	Wait-list control	Increased self-esteem (Self-Esteem Inventory), coping (Birmingham Coping Inventory), and likelihood of assertiveness (Assertion Inventory) at 3, 6, and 18 months follow-up.	High
Dittrich (1984) *USA*	*n* = 23 wives of alcoholics	Group intervention; 8 sessions (and optional)	Wait-list control	Improved self-concept (Tennessee self-concept scale); decreased anxiety (Taylor Manifest Anxiety Scale) and enabling behaviour Memphis Enabling Behaviors Inventory (MEBI) post treatment and 6 months later.	Unclear
Kim (2014) Korea	*n* = 29; wives of alcoholics	Forgiveness therapy; 12 sessions	Standard care *	Improved resilience and self-esteem at 3-month follow-up.	Low
Osilla (2018); Rodriguez (2018); *USA*	*n* = 306; partners of heavy drinking service members and veterans; 95% female	Partners Connect (web-based intervention) *	Wait-list control	Lower anxiety (General Anxiety Scale—GAD-7), depression (Patient Health Questionnaire—PHQ-9), and increased social support (Medical Outcome Study survey). NS relationship quality (Quality of Marriage Index—QMI), anger (State-Trait Anger Expression Inventory—STAXI), or family conflict (Family Environment Scale-FES) at 5 months follow-up.	Low
Rubio (2013) *Spain*	*n* = 188; mainly wives (90.4%) of alcohol misusers	Family self-help; 10 sessions	Standard care	Improvements in somatization, depression, anxiety, and phobia as measured on Symptom Checklist—90 (SCL-90), mental health and social role as measured on short form—36 (SF-36) at 6 months follow-up.	Low
Rychtarik (2005) *USA*	*n* = 36; female partners of alcohol misusers	(1) Coping skills training (CST); (2) Al-anon *	Wait-list control	CST and Al-anon reduced depression on Beck Depression Inventory (BDI-1A). CTS lower incident of partner violence (self-report) at 12 months follow-up.	Low
Zetterlind (2001) Hansson (2004) *Sweden*	*n* = 39; partners of alcoholics; 92.3% female	(1) Coping skills (individual) 5 sessions; (2) CBT (group) 13 sessions	Standard information session; 1 session	NS on coping behaviour scale; the SCL-90, global severity index, Alcohol Use Disorder Identification Test (AUDIT) (affected other alc use) at 12 months follow-up.	Low
*Interventions for families of an adult who uses drugs*					
Bortolon (2017) *Brazil*	*n* = 335, 88.7% female; mostly mothers (62%) of drug users	Tele-intervention; 9 telephone sessions	Bibliotherapy	Twice as likely to modify their co-dependent behaviour (Holyoake co-dependency index—HCI) at 6 months follow-up.	Low
Faghih (2019) *Iran*	*n* = 64; family members of drug dependent relative; 78% female	CBT; 16 sessions	No intervention	Reduced burden of care as measured on Zarit Burden Scale at 3 months follow-up.	Low
*Interventions for families of an adult who uses alcohol and/or drugs*					
Haddock (2003) *UK*	*n* = 36; mostly parents (66.6%) and females (75%) of dually diagnosed relatives	CBT; 29 sessions	Standard care *	NS on physical and mental health (GHQ, BDI), Social Behaviour Assessment Schedule (SBAS), and Relatives Cardinal Needs Schedule (RCNS) at 12 months follow-up.	Low

* Number of sessions not reported in paper.

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
