# Peer review of "Psychosocial Interventions to Improve Psychological, Social and Physical Wellbeing in Family Members Affected by an Adult Relative’s Substance Use: A Systematic Search and Review of the Evidence"

_ijerph, 2021, doi:10.3390/ijerph18041793_

Round 1
Reviewer 1 Report
Strengths of this paper include:
This review covers a very important and unstudied area in the field of substance abuse research.
- Wide breadth of databases covered.
- Good discussion and coverage of limitations. To improve clarity and readability, please address the following points:
1. Structure of sections:
- Parent and child interventions: it was hard in this section to follow which studies had an intervention on which family members, which family members were being assessed, and what outcomes were being evaluated. The first sentence of each paragraph in this section should more clearly define that, and the last sentence should effectively recap the subsection.
- It should be clarified whether each study is looking at a particular type of substance use (i.e. alcohol or opioids) or collectively looking at multiple substance use disorders.
- Readers may be unfamiliar with the pharmacotherapy names (i.e. methadone) and instead, the paper should clarify that in a study of individuals on methadone treatment for opioid use disorder. 2. General comments:
The paper has multiple instances in which abbreviations without first defining them, both in tables and the text.
- Recommend slightly more description about the different interventions.
- Many sentences need commas to better define lists, and clauses should should be the same in format (i.e. all nouns, or all short phrases).
- [Substance] use disorder is the accepted terminology according to the DSM-5. Recommend systematically updating the paper to remove references to alcohol abuse, etc.
- Table 2 is hard to follow
- recommend reformatting column widths and symbols.
- It is unclear what is meant in line 423: Structural subordination deepens the strain of a relatives substance use.
Author Response
Reviewer 1 Comment |
Response |
Structure of sections |
|
Parent and child interventions: it was hard in this section to follow which studies had an intervention on which family members, which family members were being assessed, and what outcomes were being evaluated. The first sentence of each paragraph in this section should more clearly define that, and the last sentence should effectively recap the subsection. |
We have re-organised the section addressing parent and child interventions; further separating the narrative synthesis by who received the intervention and the outcomes that were being assessed. |
It should be clarified whether each study is looking at a particular type of substance use (i.e. alcohol or opioids) or collectively looking at multiple substance use disorders. |
We have added this detail into the narrative synthesis throughout and organised table 1 within each of the intervention types, by substance type (alcohol, drugs, alcohol or drugs) |
Readers may be unfamiliar with the pharmacotherapy names (i.e. methadone) and instead, the paper should clarify that in a study of individuals on methadone treatment for opioid use disorder. |
This clarification has been added to the first reference to methadone on line 173-174 |
General comments |
|
The paper has multiple instances in which abbreviations without first defining them, both in tables and the text. |
Definitions added within text prior to using abbreviations and in table a key to abbreviations relating to outcome measures is provided at the end. |
Recommend slightly more description about the different interventions. |
Detail added to each intervention section as follows:
Parent and child interventions: ‘There were broadly two types of interventions for parents and children affected by parental substance use; those that intervened with the parent to enhance parent skill and those that intervened with the child to address the impact that their parent’s substance use has had upon them. We identified five trials which intervened with the substance using parent to enhance their parenting skill and family functioning. In these trials the affected non-using partner and/or child were typically involved to support change within the substance using parent’s behaviour, for example, to provide a means to practice the newly acquired skill, or to reinforce positive behaviour.’
BCT: ‘Within the conjoint sessions, couples were typically encouraged to discuss how the non-using parent can positively support the using partner; are taught how to communicate more effectively and increase positive behavioural exchanges.’
Systemic family therapy: ‘Systemic family therapy typically treats the substance users and their family as a system; recognizing the role of the family in the development and treatment of the adult relative’s substance use. It is generally assumed that positive changes in the family system will result in positive changes in the relative’s substance use.’
Unilateral family interventions: ‘Unilateral family interventions intervene with the affected family member in order to teach them strategies by which they can support or influence the substance using family member to address their substance use. The interventions typically include education on the process of change, on how to effectively place pressure upon substance user to action change as well as support to address the affected family member’s behaviour which may enable the relative’s substance use.’
Psychosocial interventions for the individual family member: ‘Psychosocial interventions for the family members of a substance user are non-pharmacological therapeutic interventions delivered to individuals or groups, which seek to tackle the psychological, social and personal problems experienced by the affected family member.’ |
Many sentences need commas to better define lists, and clauses should should be the same in format (i.e. all nouns, or all short phrases) |
The sentence structure has been addressed throughout. |
[Substance] use disorder is the accepted terminology according to the DSM-5. Recommend systematically updating the paper to remove references to alcohol abuse, etc. |
This has been changed throughout the review |
Table 2 is hard to follow. recommend reformatting column widths and symbols.
|
We have re-designed table 2 to more effectively communicate the available evidence as an evidence and gap map. We have also repositioned it in the paper (it is now table 10 to enable better introduction and explanation of its purpose. |
It is unclear what is meant in line 423: Structural subordination deepens the strain of a relatives substance use. |
Clarification has been added: ‘Structural subordination and dependence, wherein family members are reliant financially or socially upon the substance using relative, deepens the strain of a relatives substance use (92).’ |
Reviewer 2 Report
Although the topic of this study really is of interest to the field and relevant for clinicians, I have serious doubts on the scientific value as it is presented in its current format. Following are my main concerns:
- The strength and both weakness of the manuscript is it broadness. I do applaud the authors of reading and examining so many articles, but the very broad inclusion results in so much, fragmented, information that it is very hard to present all the needed information and draw some clear conclusions.
- In the search strategy the search-terms need to be specified
- Unless I am missing something, but I see only one table, including only four studies (table 1).
- Much more information is needed to substantiate the conclusions as synthesised by the authors in table 2. More specific, we would need tables, including all studies (65), reporting on: the sample, design used, outcome measures used, moment of measurement, results and statistic data per study. I do realise that this is an enormous amount of work, but otherwise we only can believe the author's conclusions on their word, but the reader can not verify.
- It is unclear whether the interventions explored are stand alone or on-top of TAU for substance use disorder for the addicted patient. This relation can confound any other effect.
- The choice to divide the results section by type of intervention is defendable, but for many clinicians results according to substances of abuse might be very relevant, e.g. from family perspective there are often big differences between alcohol, cannabis or stimulant problems;
Taken together a lot more work and information is needed to be presented in the paper, to allow for the conclusions suggested in the discussion and conclusion. An alternative approach could be to narrow done the search, e.g. with a focus on social aspects (i.e. relational quality, aggression, ..) allowing for a more in depth work out of the findings.
Author Response
Reviewer 2 Comment |
Response |
· In the search strategy the search-terms need to be specified
|
We have added key search terms within the methods section and also uploaded the full search strategy as a supplementary file |
· Unless I am missing something, but I see only one table, including only four studies (table 1). |
Please accept our apologies. The large size of this table appears to have caused difficulties whilst the PDF was being formatted. We have uploaded this table and repositioned it within the paper to come at the end and therefore avoid these formatting difficulties. This table is now ‘table 2’ |
· Much more information is needed to substantiate the conclusions as synthesised by the authors in table 2. More specific, we would need tables, including all studies (65), reporting on: the sample, design used, outcome measures used, moment of measurement, results and statistic data per study. I do realise that this is an enormous amount of work, but otherwise we only can believe the author's conclusions on their word, but the reader can not verify.
|
As above, details of the sample, design, outcome measure and findings were included in what we previously referred to as ‘table 1’ (unfortunately not previously visible to the reviewer). We have also added the moment of measurement to this table. As we have repositioned this table in the paper, we now refer to this as ‘table 2’. |
· It is unclear whether the interventions explored are stand alone or on-top of TAU for substance use disorder for the addicted patient. This relation can confound any other effect.
|
We have added further detail to our description of studies section and also reported comparison conditions within table 2. |
· The choice to divide the results section by type of intervention is defendable, but for many clinicians results according to substances of abuse might be very relevant, e.g. from family perspective there are often big differences between alcohol, cannabis or stimulant problems;
|
We have grouped the summary of findings in table 2 by substance (alcohol, drugs or alcohol and/or drugs). We have also included details of the substance used by the adult relative within the narrative synthesis. |
Round 2
Reviewer 2 Report
I thank the authors for their extensive work elaborating on the important Table 2 - this gives ample information supporting the article's content
Author Response
We thank the review for their encouraging response. We have responded to all comments of the academic editor as follows:
Line 201 - this sentence has been altered to communicate more clearly that systems therapy did not improve the outcomes significantly more than the comparison condition (problem solving approach).
line 209 - 'to' removed
line 222 - line removed
line 273 - and has been changed to 'or'
Line 479 - positive has been changed to 'positively'
Line 496 - where has been changed to 'were'
line 501 - the point we are attempting to make is that these interventions have been shown to be effective in other populations. We have reworded to make this clearer to the reader: "These interventions were typically based upon efficacious interventions which have previously been found to be effective in general populations for depression and anxiety such as cognitive behavioural therapy (91),"
line 515 to 519 - the content of these sentences are the findings of the review. As they represent our original contribution (e.g. that the trials of conjoint therapy which focused on the substance user did not bring about mental health benefit in the affected other), they do not have references to attach.
Line 565 - reference added
Table 2 - risk of bias summary added